# Geometry and Kinematics of the Central Fault Zone, Fula Sag, Central Africa Shear Zone

**Yanqi Wang, Guangya Zhang \*, Guoqi Wei \*, Zhuxin Chen, Rong Ren, Yuqing Zhang and Ke Geng**

Research Institute of Petroleum Exploration and Development (RIPED), PetroChina, Beijing 100083, China; wangyq7@petrochina.com.cn (Y.W.); chenzhuxin@petrochina.com.cn (Z.C.); rr@petrochina.com.cn (R.R.); yqzhangzj@126.com (Y.Z.); gengkkee@163.com (K.G.)
\* Correspondence: zhanggy2022@126.com (G.Z.); weigq@petrochina.com.cn (G.W.)

**Abstract:** The Central Africa Shear Zone (CASZ) harbors abundant hydrocarbon resources within its Central Fault Zones (CFZs). The studies of CASZ have dominantly focused on the evolution and superimposition processes of prototype basins in CASZ. Meanwhile, research on the geometry and segmental growth of main faults in CFZs remains poorly understudied, which limits hydrocarbon exploration. In this paper, we focus on the CFZ of the Fula sag as an example of CASZ and utilize the 3-D throw mapping technique along with the maximum throw subtraction method to investigate its geometric and growth processes. Results show faults in the northern and central parts of the CFZ form multiple Y-shaped combinations, and a system of sub-parallel faults in the south forms the bookshelf faults. Meanwhile, the divergent overlapping transfer zone is identified in the CFZ. Our investigation found abrupt changes in throw-distance diagrams of main faults in the CFZ, which indicate that the main faults, $F_1$, $F_3$ and $F_2$, are laterally segmented into 4, 4, and 3 segments, respectively. As an intracontinental passive rift basin, the Fula sag has undergone three major rifting cycles since the Early Cretaceous, triggered by the segmental expansion of the Atlantic Ocean, the rapid opening of the Indian Ocean, and the separation of the Red Sea. Our analysis also reveals that the main faults in the CFZ were primarily active during the second rifting, with the fault segments undergoing isolated growth, soft linkage, and eventually forming fully grown faults during the third rifting. We observe a significant decrease in activity intensity during the transition between the second and third rifting cycles. Our findings provide insights into the growth and activity of the CFZ faults, which are applicable to other CFZs of similar origin in rift basins, and provide suggestions for hydrocarbon exploration and production.

**Keywords:** segmented growth of faults; 3-D throw mapping technique; maximum throw subtraction method; Fula sag; intracontinental passive rift basin; Central Africa Shear Zone

## 1. Introduction

As an important tectonic zone for oil and gas exploration, the central fault zone (CFZ) of the rift basin is the focus of intensive research due to its proximity to the hydrocarbon generation center and the possibility of obtaining oil and gas charging nearby. The overlapping area of the normal fault segments in the CFZ is where the relay ramp develops and has a significant control on CFZ physiography [1], sediment distribution [2,3], fluid migration pathways [4,5], and hydrocarbon accumulation. For example, the CFZs of the Dongying depression [6–9] and the Huimin depression [10–12] in the Bohai Bay basin and the Xihu depression in the South China Sea [13,14] are high-quality and high-yielding hydrocarbon enrichment zones. A series of intracontinental multiphase passive rifts are distributed along the Central Africa Shear Zone (CASZ), of which the CFZs are rich in hydrocarbon resources and have now entered the stage of lithologic reservoir exploration [15,16]. In the presence of abundant hydrocarbon sources, the key to exploring lithological reservoirs lies in identifying the distribution of favorable sand bodies, specifically the locations where

relay ramps develop during the growth process of faults. Therefore, studying the geometry and the segmental growth of the main faults on the CFZs in CASZ is helpful to predict the locations of lithologic reservoirs, providing a reference for later hydrocarbon exploration.

Several studies have discussed the geological model of rift basins in the CASZ [17–20]. The analysis demonstrates that basins in CASZ have roughly experienced three phases of rifting cycles since the Early Cretaceous due to the Atlantic opening, the rapid opening of the Indian Ocean, and the opening of the Red Sea rift. Some studies also address the geometry of the present-day CFZs in CASZ [21–26], which provides limited information on the segmental growth process of the normal faults and is unable to restore the kinematics. For instance, some interpreted the CFZ in Fula sag, middle CASZ as the synthetic approaching or convergent approaching transfer zone [24,25] according to Morley's classification of the transfer zones [27]. However, others argued that it exhibits characteristics of the convergent overlapping transfer zone [26].

Various models of normal fault evolution have been proposed in the literature, involving a progression from underlapping to overlapping fault geometries over time [28–35]. These models suggest that faults grow by lengthening their segments until they eventually overlap and become spatially closer, leading to fault linkage. The growth of fault segments involves different stages, including isolated growth, soft linkage (where fault surfaces are isolated but connected by ductile strain in the rock volume between them), and hard linkage (where fault surfaces are joined) [29,36]. Characteristically, there is variability in throw-distance profiles for fault segments and linked faults as they interact and undergo linkage. Throw transfer through relay ramps results in steep throw gradients along fault segments at oversteps [28].

Currently, there are two main methods for throw and segmentation growth recovery of faults that are able to restore the segmental growth process of main faults in the CFZs of CASZ: the vertical subtraction method [36–38] and the maximum throw subtraction method [39]. The vertical subtraction method, however, is applicable only to the "throw accumulation—length fixed" fault growth mode. For tectonic-genetic faults, the maximum throw during fault segment growth increases linearly with the extension length [40–43], so the maximum throw subtraction method can truly reflect the evolutionary history of fault segment growth.

This paper uses the 3-D throw mapping technique and the maximum throw subtraction method to present a detailed description and analysis of the CFZ on the Fula sag and middle CASZ. The aim is to demonstrate a comprehensive understanding of the geometrical and kinematic evolution of an extensional CFZ in the intracontinental passive rift basin based on well-documented growth strata that constrain detailed spatial and temporal fold-fault development.

## 2. Geological Setting

The basins in the CASZ, namely the Central African Rift System (CARS, Figure 1a), are Mesozoic-Cenozoic intracontinental passive rift basins [44]. They experienced three rifting cycles, driven by the expansion of the Atlantic Ocean, the northward movement of the Indian Plate, and the openness of the Red Sea. The first rifting was primarily influenced by the Atlantic opening, resulting in a right-lateral strike-slip CASZ [45–48]. As the shear movement extended northeastward from the Congo Craton, the CASZ transformed from a shear stress field to an extensional stress field [21]. The second rifting was mainly influenced by the rapid opening of the Indian Ocean, causing a weakening near-NE-striking stress field in central Africa [49]. The third rifting was associated with the separation of the Red Sea, leading to an east-to-west weakening and near-NE-striking extensional stress field within the African plate [50,51]. Furthermore, they also exhibit characteristics of a pull-apart basin, which are evident in the distribution pattern of major highs and lows, the shifting of the depocenter over time, the thermal regime of the basin, and the formation of traps [52–54].

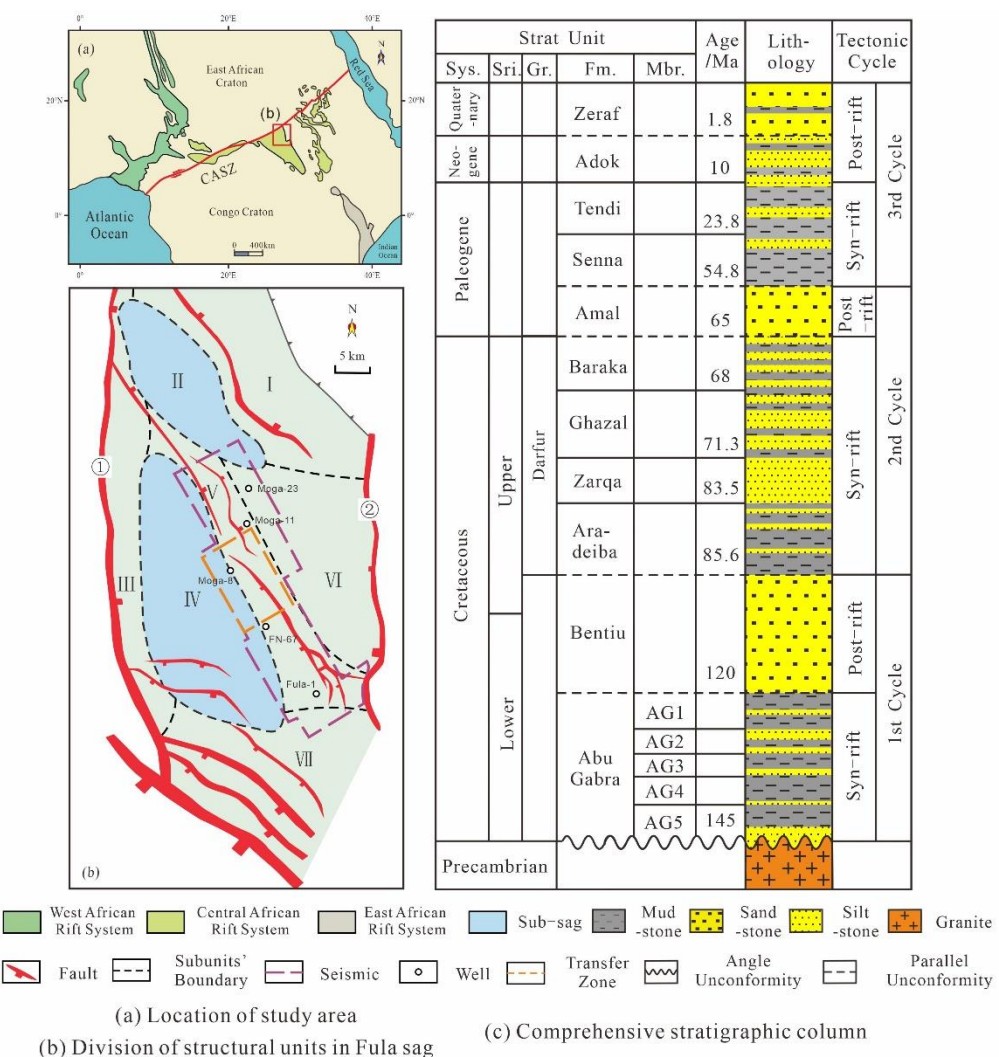

(a) Location of study area

(b) Division of structural units in Fula sag

(c) Comprehensive stratigraphic column

**Figure 1.** (**a**) Location of study area, (**b**) division of structural units in Fula sag, and (**c**) comprehensive stratigraphic column of Fula Sag (Modified after [20,23,46,55]). Noted that (**a**) The Fula sag, as a part of the Muglad Basin, tectonically situated in the middle CASZ. (**b**) The CFZ of the Fula sag have three main faults with similar NWW strike. (**b**) Since its formation in the Early Cretaceous, the Fula sag has experienced three phases of rifting cycles, each of which consists of a syn-rift as well as post-rift thermal subsidence. Tectonic units: I–VII. are the northeastern fault-array-and-ramp zone, northern sub-sag, west-ern steep-slope zone, southern sub-sag, CFZ, eastern ramp zone, and southwestern fault-array-and-ramp zone, respectively; ① and ② are boundary faults of the Fula sag; the CFZ faults depicted in (**b**) represent F1, F3, and F2 from south to north. CASZ = Central Africa Shear Zone.

The Fula sag, as a part of the Muglad Basin, is located in the middle portion of the CASZ, covering an area of about 5000 km². The sag as a whole has sub-N-S longitudinal spreading, while the internal structure is NW-SE trending. Based on the sag structure and fault characteristics, the Fula sag can be divided into seven sub-tectonic units from north to south, including the northeastern fault-array-and-ramp zone, the northern sub-sag, western steep-slope zone, the southern sub-sag, the CFZ, the eastern ramp zone, and the southwestern fault-array-and-ramp zone (Figure 1b) [55]. The Fula sag is a graben controlled by two boundary faults (① and ② in Figure 1b) in the east and west, but fault ① is more active, and the western deposits are thicker, with a thickness reaching up to 11,400 m. The sag as a whole is an "asymmetric graben", with thicker strata in the west.

The main faults of the CFZ have an overall NNW strike with a diagonal distribution, and the basement of the CFZ rises upward, separating the depocenter of Fula sag into two, namely the northern sub-sag and the southern sub-sag. There are three main faults here,

named $F_1$, $F_3$, and $F_2$, from south to north (Figure 1b). The dip of the northern two main faults is eastward, while the dip of the southern one is westward.

Since its formation in the Early Cretaceous, the Fula sag has been influenced by tectonic events such as the Atlantic opening, the rapid opening of the Indian Ocean, and the opening of the Red Sea rift, forming a large number of high-angle normal faults and experiencing three phases of rifting cycles [17–20]. Each rifting phase consists of a syn-rift as well as post-rift thermal subsidence (Figure 1c). The Abu Gabra Formation (referred to as the AG Formation) and the Bentiu Formation constitute the syn-rift phase and post-rift phase of the first rifting, respectively. The AG Formation was deposited from 145 Ma and belongs to the Lower Cretaceous, which can be subdivided into $AG_1$–$AG_5$ Members from top to bottom [56]; the Bentiu Formation was deposited from 120 Ma and includes the strata from the top of the Lower Cretaceous to the bottom of the Upper Cretaceous. The sedimentary period of the Darfur Group and the Amal Formation are the syn-rift stage and the post-rift stage of the second rifting, respectively. The Darfur Group was deposited from 85.6 Ma and belongs to the Upper Cretaceous; the Amal Formation was deposited from 65 Ma and belongs to the Paleocene. The sedimentary period of the Senna-Tendi Formation and Adok-Zeraf Formation are the syn-rift stage and the post-rift stage of the third rifting, respectively. The Senna and the Tendi Formations were deposited from 54.8 Ma and 23.8 Ma, respectively, including the Eocene-Miocene stratigraphy.

## 3. Datasets and Methodologies

The seismic data utilized in this study comprise two 3D seismic surveys, namely the Fula and the Moga 3D seismic surveys, which were acquired between 2000 and 2003. Covering an area of 500 $km^2$, the surveys recorded a length of 5.0 s two-way travel time (TWT), with a line spacing of 9.375 m × 9.375 m and a vertical sample rate of 2 ms. The polarity of the seismic data is positive. The dominant frequency ranges from 30 to 50 Hz, and the acoustic velocities of the AG to Zaraf section range between 2000 and 5000 m/s, resulting in an estimated vertical seismic resolution of 40–167 m. Stratigraphic correlation was performed using wells Fula-1, FN-67, Moga-8, Moga-11, and Moga-23 (Figure 1b), among more than 30 exploration and evaluation wells in the study area.

Five key horizons, namely the top of the Amal Formation, the top of the Darfur Group, the top of the Bentiu Formation, the top of the $AG_1$ Member, and the top of the $AG_2$ Member, were interpreted across the research area to illustrate the lateral and vertical variations in the structure of the extensional faults (Figure 2). The fault system under investigation was labeled $F_1$ to $F_{13}$ (Figure 2). Based on the seismic interpretation results, an accurate description of the geometry for the Fula-Moga transfer zone is provided.

After interpreting the horizons and faults, the 3-D fault planes were created and visualized using corner point gridding in the Structural Modeling module of PETREL 2020.2 software. Subsequently, the throw attribute of the main fault planes was generated in PETREL software using the 3-D throw mapping technique described by Lister [57] and applied by Dutton, et al. [58]. The throw represents the vertical component of the dip-separation of a normal or reverse fault, measured in a vertical cross-section perpendicular to the strike of the fault [59]. Throw-distance graphs (T-X) for individual main faults $F_1$, $F_3$, and $F_2$, associated with the CFZ, were generated every 400 m along strike using the exported throw data from the software. Finally, the maximum throw subtraction method was employed to quantitatively study the growth process of each fault segment and reconstruct the fault segmentation growth of the entire fault zone.

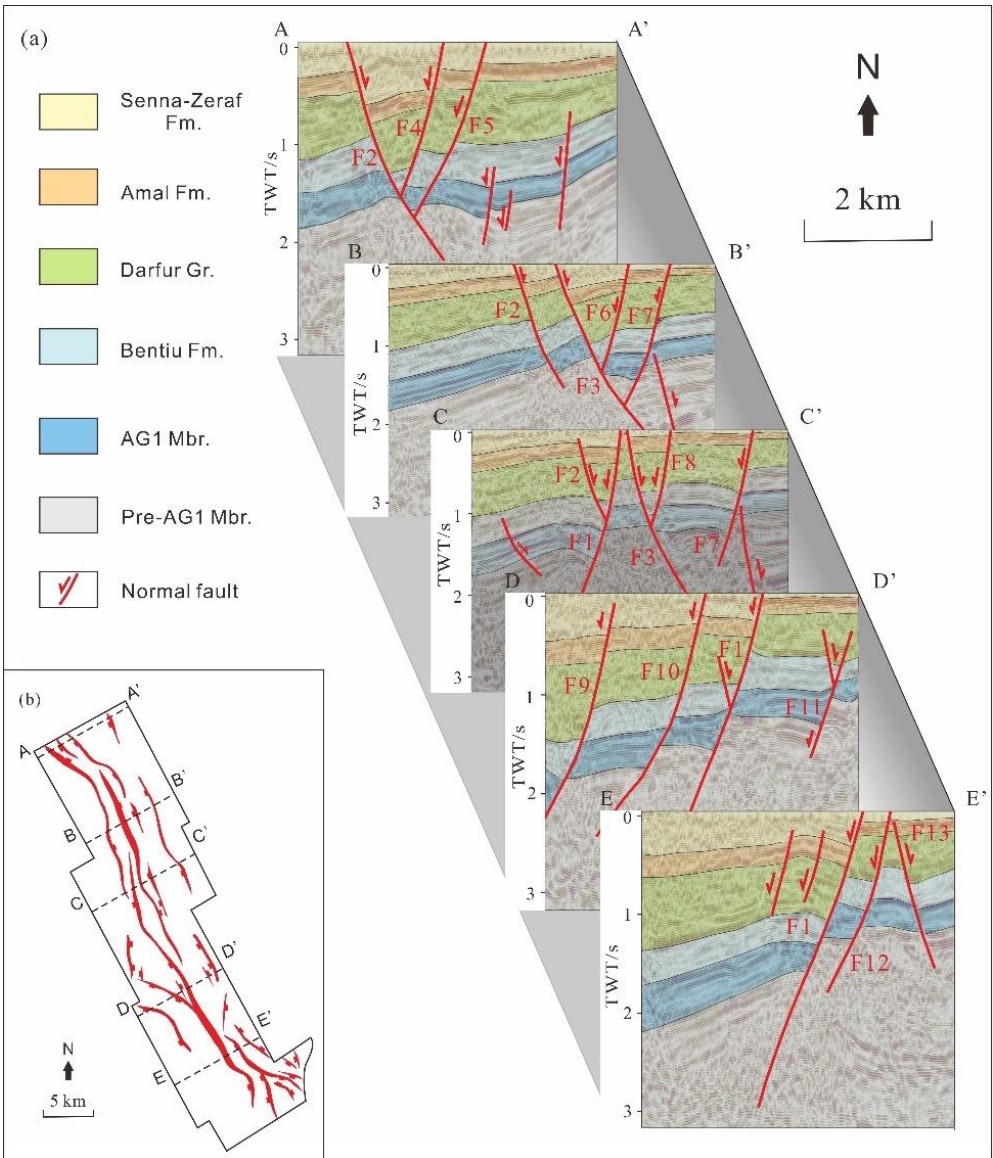

**Figure 2.** Seismic profiles A–A′ to E–E′ (**a**) across the CFZ and their location (**b**) on the Top Bentiu Fm. The CFZ is predominantly featured by NW-striking faults, with west-dipping faults in the south and vice versa in the north. The secondary faults in the north combine with fault $F_2$ or fault $F_3$ to form a Y-shaped structure. Main faults in the central area form multiple Y-shaped combinations. A system of sub-parallel faults in the south forms the bookshelf faults. TWT = Two-way travel time.

## 4. Structural Styles

Based on the drilling, logging, and seismic data of the CFZ, we calibrated synthetic seismic records, interpreted seismic data, and compiled tectonic maps for the study area, which laid the foundation for our investigation into the spatial distribution characteristics of the faults and the three-dimensional morphology of the transfer zone.

### 4.1. Fault Distribution Characteristics

Restricted by the low quality of seismic data, the seismic isotropic axis in this 3D area between the top of the $AG_3$ Member and the basement is poorly identified. This study focuses on the interpretation of five layers, including the top of the Amal Formation, the top of the Darfur Group, the top of the Bentiu Formation, the top of the $AG_1$ Member, and the top of the $AG_2$ Member.

The A-A′ section of Figure 2a is located in the northern part of the study area, and the stratigraphy dips westward as a whole, with a syncline-type structure in the east-dipping to the northern sub-sag. The main fault $F_2$ with smaller antithetic faults develops a Y-shaped structure. The syncline-like structure on the east may have been formed by a rollover due to the extensional fault propagation fold mechanism, which could have resulted from either trishear or flexural slip along faults. The fault distribution map of the top of the Bentiu Formation (Figure 2b) also shows that there are several locations where the strike of the main faults dramatically changes, and it is thought that the main faults may be segmented into several fault segments in these locations.

Located in the north of the study area, the B-B′ section crosses faults $F_2$, $F_3$, $F_6$, and $F_7$ from west to east, respectively. For this section's close proximity to the tip of fault $F_2$, it has a smaller throw in this section. $F_3$, however, has a larger throw since this section is close to its depositional center. The fault is a listric fault, and on the hanging wall of this fault, the Darfur Group and overlying strata exhibit considerable thickness, indicating the activity of this fault during the second rifting and third rifting. The fault strike here is all NNW. Fault $F_2$ and $F_3$ dip to NEE in the form of a fault array. Fault $F_6$ and $F_7$ combine with $F_3$ to form a Y-shaped structure. As seen in the B-B′ section of Figure 2, the stratigraphic depth deepens rapidly to the east in front of faults $F_2$ and $F_3$, indicating the existence of a local depositional center near each of the two faults.

The C-C′ section is in the middle of the study area, the transition area between the east-trending faults in the north and the west-trending faults in the south, and also the area where the Fula-Moga transfer zone (indicated by the orange dashed line in Figure 1a) is developed. The profile crosses faults $F_2$, $F_1$, $F_3$, $F_8$, and $F_7$ in sequence from west to east. The central part is a horst controlled by faults $F_1$ and $F_3$, while faults $F_1$ and $F_3$ form a graben with fault segments outwardly, thus forming a "graben-horst" combination in the profile. Fault $F_1$, fault $F_3$, and fault segments generate a Y-shaped combination, and the stratum here incrementally evolves into two broad and gentle anticlines on both sides. The eastern anticline is controlled by a west-dipping fault.

The D-D′ section is located in the south of the study area crossing the west-dipping faults $F_9$, $F_{10}$, and $F_1$ from west to east. These three faults emerge as a faulted terrace, with the strike progressively shifting from NNW to NW, and fault $F_{10}$ eventually terminates at fault $F_1$ while spreading to the south. As can be seen from this section, the stratigraphy appears to be relatively gentle on the west of fault $F_1$ and accumulatively dips to the southern sub-sag, suggesting that the regional deposition center of this area is in the southern sub-sag and the deposition process is controlled by the faults at the same time.

The E-E′ section is located in the southern part of the study area, and the section crosses the west-dipping faults $F_1$ and $F_{12}$, and the east-dipping fault $F_{13}$ from west to east. Fault $F_1$ has the largest throw and is also the most active. Several fault segments with the same dip are developed in the shallow layer to the west of fault $F_1$. An enormous roll-over is developed on the west of the fault $F_1$, and its axial surface is parallel to the plane of fault $F_1$. A horst forms in the middle of this section as the result of the combined effect of faults $F_{12}$ and $F_{13}$.

In a nutshell, the CFZ is predominantly featured by NW-striking faults, with west-dipping faults in the south and east-dipping faults in the north. The fault segments in the north combine with fault $F_2$ or fault $F_3$ to form a Y-shaped structure. Main faults in the central area form multiple Y-shaped combinations and enable the stratum to develop a "graben-horst" combination. A system of sub-parallel faults in the south forms the bookshelf faults. The stratigraphy in the CFZ generally dips to the west, and a clear syncline can be observed in the northern area, while a roll-over parallel to fault $F_1$ can be detected.

### 4.2. 3D Visualization of the Transfer Zone

The fault striking of the CFZ is predominantly NW, with an overall diagonal spreading. The southern section is a west-dipping fault array, and the main fault is fault $F_1$. The

northern section is an east-dipping fault array, with fault $F_2$ and fault $F_3$ being the main faults. The Fula-Moga transfer zone is located in the middle section of the CFZ and is in the transition area between fault $F_1$ and fault $F_3$ in map view. These normal faults overstep in map view and have a divergent dip direction. The middle section appears to be a horst in tectonic style, and the transfer zone gradually transits to the north and south in the form of a breached relay ramp. The Fula-Moga transfer zone, therefore, can be considered the divergent overlapping type (Figure 3), based on the classification scheme of Chen, et al. [60]. A handful of fault segments are developed on the horst, and their strikes are either parallel or oblique to those main faults. In the northwest, two small anticlines can be observed that are nearly perpendicular to the main fault.

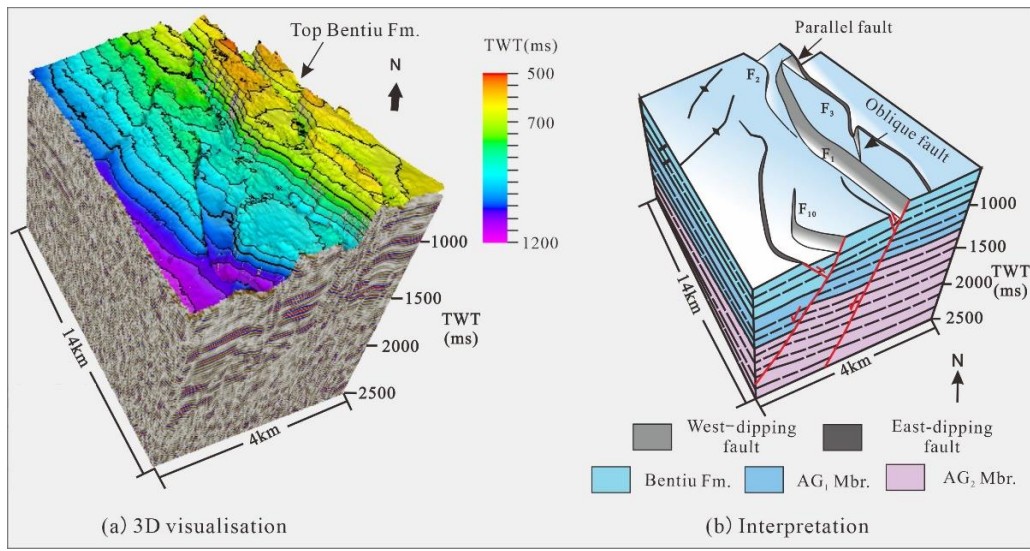

**Figure 3.** 3D visualization of the seismic geobody (**a**) and interpretation (**b**) showing detailed fault structures in the Fula-Moga transfer zone. Its location is shown in Figure 1. The main faults, $F_1$ and $F_3$, overstep in map view and that have the divergent dip direction. The Fula-Moga transfer zone is located in the middle CFZ. It appears to be a horst, and gradually transits to the north and south as a breached relay ramp.

## 5. Fault Throw Analysis

After the horizons and faults were interpreted, the 3-D fault plane and the throw attribute (Figure 4) could be created and displayed by corner point gridding in the Structural Modeling module, PETREL software, which shows the throw variation in the main faults of the CFZ, with the color contours on the fault plane corresponding to the throw values. This can also give us a better understanding of the segmental characteristics of the entire fault plane.

Fault $F_1$ (Figure 4a) has four local throw maximum points distributed on the fault plane, indicating that the fault is connected by four fault segments, among which the fault throw contours of faults $F_{1-1}$, $F_{1-2}$, and $F_{1-3}$ on the south are roughly fused into one. The darkest color appears at fault $F_{1-2}$, which indicates that fault $F_{1-2}$ is the most active among three fault segments. The local throw maximum points of all four fault segments occur at the top of the Bentiu Formation, suggesting that all four fault segments have been active since the Darfur period. Fault $F_3$ (Figure 4b) has four local throw maximum points distributed on the fault plane, indicating that the fault is segmented into four fault segments, among which the fault throw contours of $F_{3-2}$, $F_{3-3}$, and $F_{3-4}$ on the north side are entirely fused into one. The darkest color at fault $F_{3-3}$ indicates that fault $F_{3-3}$ is the most active among four fault segments. The local throw maximum points of all four fault segments occur at the top of the Bentiu Formation, suggesting that all four fault segments have been active since the Darfur period. Similarly, it can be seen that fault $F_2$ (Figure 4c) is composed of two larger faults $F_{2-1}$ and $F_{2-3}$ and connected to a smaller fault $F_{2-2}$.

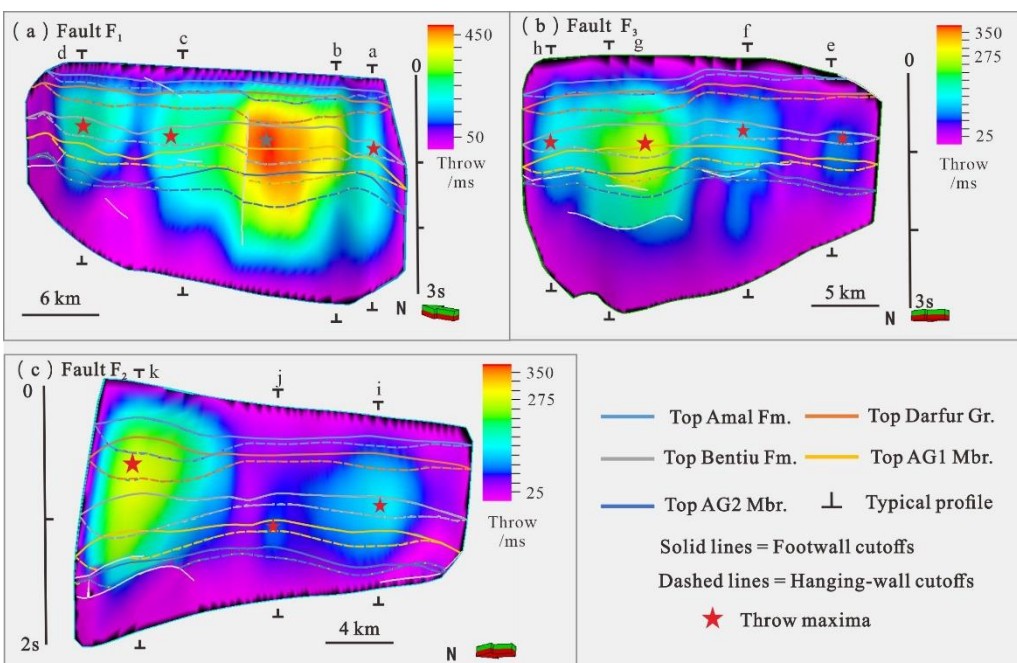

**Figure 4.** Three-dimensional visualization of the main faults of the CFZ with the fault surfaces color contoured to correspond to values of fault throw. These diagrams record the horizon-offset geometry for five mapped horizons and display the spatial variation of throw across the fault surface. Hanging-wall cutoff lines are solid, whereas footwall cutoff lines are dashed. Note that the main faults of CFZ, $F_1$, $F_3$ and $F_2$, exhibit 4, 3, and 3 throw maxima on their fault planes, respectively. The labels a–k indicate the profile locations used to calculate the throw accumulation rate of the fault segments for each period.

Throw-distance graphs (T-X) for individual main faults $F_1$, $F_3$, and $F_2$ associated with the CFZ have been generated (Figure 5a–c). These plots record how fault throw varies with extension distance for a total of five layers from the top of the Amal Formation to the top of the $AG_2$ Member, which allow us to recover the segmental growth process of these faults. Throw-distance graphs allow the observation of more subtle throw variations that may be masked or overlooked.

Fault $F_1$ (Figure 5a) is approximately 30 km long, and the integral fault throw of each layer is generally characterized by a high middle and low sides. For the same layer, fault $F_1$ has four local maximum points and three local minimal points distributed from south to north, and these extreme points are evenly distributed in the longitudinal direction. The local maximum points represent the local depositional centers corresponding to the period, and the local minimal points correspond to the lateral connection points of the fault segments. Thus, it can be seen that fault $F_1$ is segmented into four fault segments, namely $F_{1-1}$, $F_{1-2}$, $F_{1-3}$, and $F_{1-4}$. Each layer sees a rapid descent in fault throw from south to north, which indicates that fault $F_{1-2}$ in the south is the most active, with the faults' activity intensity dropping dramatically to the north. For each fault segment, the throw increases from the top of the Amal Formation to the top of the Bentiu Formation and then decreases gradually. For instance, the throw of each layer of fault $F_{1-2}$ first increases from 240 ms at the top of the Amal Formation to 517 ms at the top of the Bentiu Formation and 469 ms at the top of the $AG_1$ Member, and then gradually decreases to 453 ms at the top of the $AG_2$ Member ($F_{1-2max}$ in Figure 5a), suggesting that all four fault segments that make up fault $F_1$ have been active since the Darfur period.

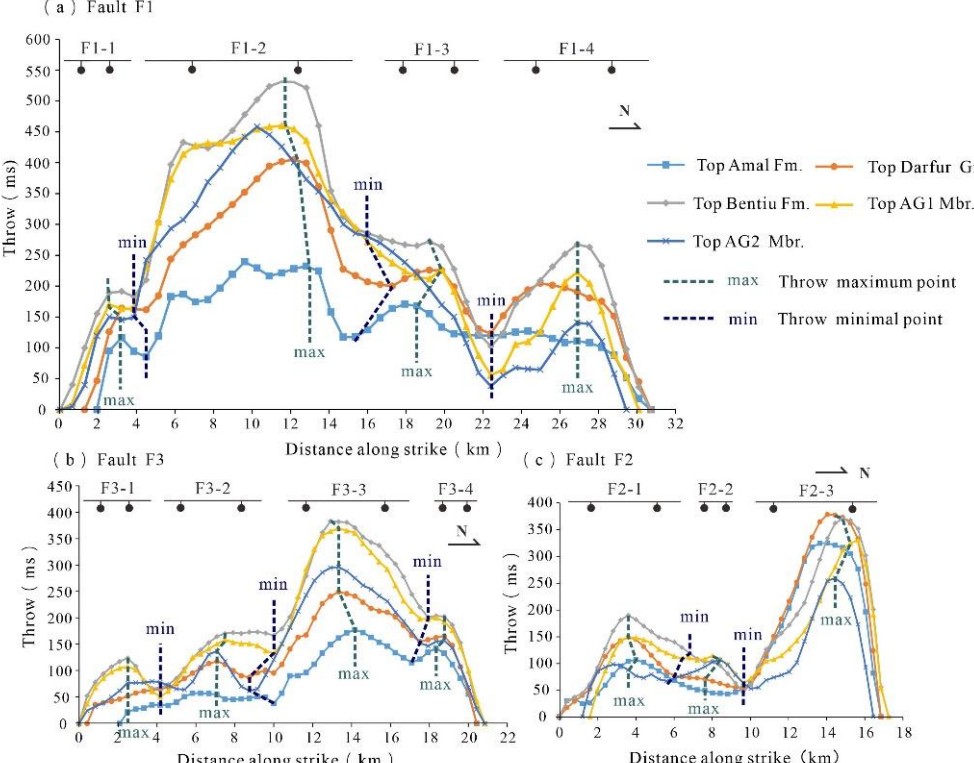

**Figure 5.** Throw-distance analysis for main faults of the CFZ. Throw profiles along present-day faults are marked by throw lows, which correspond to original regions of fault linkage and reveals the number of the fault segments. Fault $F_1$ and $F_3$ are both segmented into four fault segments, and fault $F_2$ is segmented into two larger fault segments and a smaller segment.

Fault $F_3$ (Figure 5b) is approximately 20 km long, and the fault throw is generally featured by a high middle and low sides. Each layer contains three local minimal points. From south to north, the fault throw sees a slow ascent initially and then increases sharply in two steps in the middle, and it drops rapidly after reaching its maximum value at fault $F_{3-3}$, which indicates that the fault is developed from four fault segments, namely $F_{3-1}$, $F_{3-2}$, $F_{3-3}$, and $F_{3-4}$. For each layer of fault $F_3$, the global maximum point of fault throw is located in the north of the fault, which suggests that the fault is more active in the north. The four fault segments that make up $F_3$ also show a gradual increase in the maximum fault throw from the top of the Amal Formation to the top of the Bentiu Formation, followed by a gradual decrease. For example, the throw maximum point of fault $F_{3-3}$ first increases from 178 ms at the top of Amal Formation to 377 ms at the top of Bentiu Formation and 363 ms at the top of $AG_1$ Member, and then gradually decreases to 289 ms at the top of $AG_2$ Member, indicating that all four fault segments that make up fault $F_3$ have been active since the Darfur period.

$F_2$ (Figure 5c) has an approximate length of 17 km or possibly more (as the seismic data ends before the tip of the fault in its northern portion), with each layer containing two local minimal points, which indicates that the fault plane in the CFZ is developed from three fault segments, namely $F_{2-1}$, $F_{2-2}$, and $F_{2-3}$. The global maximum point of the throw of each layer for fault $F_2$ is located in the north of the fault, indicating that fault $F_{2-3}$ in the north is more active. The throw variation pattern of each layer of fault $F_{2-1}$ in the south is similar to that of the previous faults, indicating that fault has been active since the Darfur period. The local throw maximum of fault $F_{2-2}$ increases from the top of Amal Formation to the top of $AG_1$ Member and then slowly decreases, indicating that the fault became active during the Bentiu period; the throw maximum of fault $F_{2-3}$ first rises from 306 ms at the top of the Amal Formation to 356 ms at the top of the Darfur Group and 350 ms at the top of the Bentiu Formation, and then gradually decreases to 180 ms at the top of the $AG_2$ Member,

indicating that the fault segment $F_{2-3}$ did not become active until the Amal period, a time later than the other faults.

In conclusion, none of the throw maxima of the three main faults for the CFZ is located in the middle of the fault plane, and abrupt changes in fault throw are found in all planes, which indicate that all three main faults are segmented into multiple fault segments. Among them, faults $F_1$ and $F_3$ are both segmented into four fault segments, and fault $F_2$ is segmented into two larger fault segments and a smaller fault segment. These fault segments forming the main faults of the CFZ have become active mainly since the Darfur period, and a few since the Bentiu period or Amal period.

## 6. Growth Analysis of the Main Faults

Based on the fault-plane throw analysis of the main faults in the CFZ in our previous analysis, we used the maximum throw subtraction method to quantitatively study the growth process of each fault segment so as to recover fault segmentation growth of the entire fault zone.

Fault $F_1$ is segmented into four segments, including $F_{1-1}$, $F_{1-2}$, $F_{1-3}$, and $F_{1-4}$, which have been active since the Darfur period. All four segments were in the isolated growth stage during the Darfur period (the syn-rift phase of the second rifting, Figure 6c), and the synthetic-approaching transfer zones were developed among these fault segments. During the Amal period (the post-rift of the second rifting, Figure 6b), $F_{1-1}$ and $F_{1-2}$, $F_{1-2}$ and $F_{1-3}$ started to overlap each other and entered the soft-linkage stage, and the synthetic-overlapping transfer zones were developed among these fault segments; lateral growth narrowed the distance between $F_{1-3}$ and $F_{1-4}$, but they were still in the isolated growth stage and developed a synthetic-approaching transfer zone. During the Senna-Zeraf period (the third rifting, Figure 6a), four fault segments grew rapidly, with all entering the hard-linkage stage and eventually developing a fully grown, large fault.

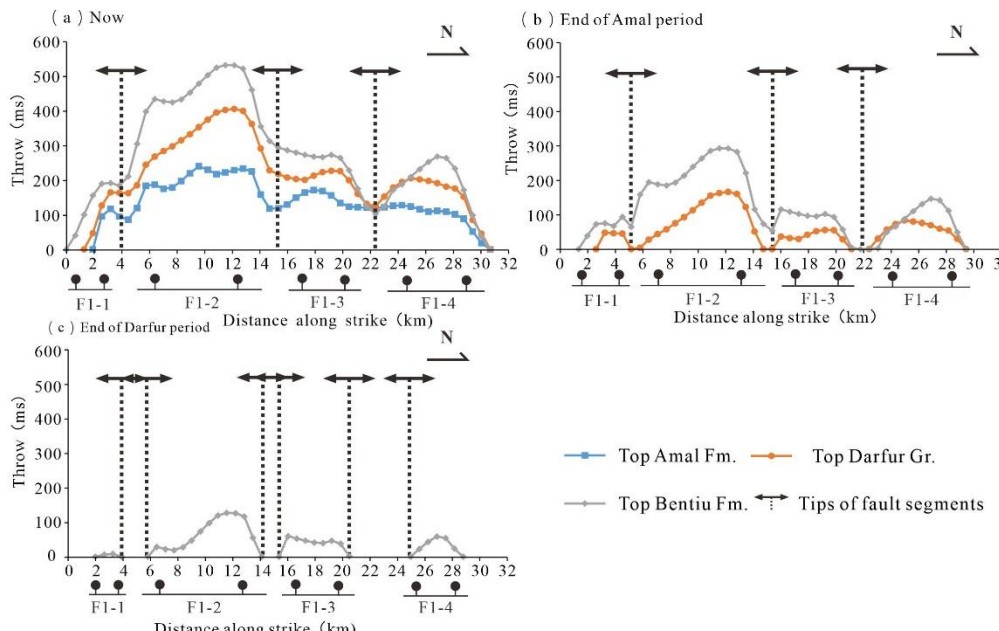

**Figure 6.** Relation between fault throw and extension distance of fault $F_1$. These plots illustrate the growth process of the fault $F_1$, indicating that the four originally separate fault segments constituting the fault $F_1$ became active during the Darfur period, entered the soft-linkage stage during the Amal period, and progressed to the hard-linkage stage during the third rifting, eventually forming a fully grown, large fault.

Fault $F_3$ is segmented into four fault segments, consisting of $F_{3-1}$, $F_{3-2}$, $F_{3-3}$, and $F_{3-4}$, all of which have been active since the Darfur period. During the Darfur period (the syn-rift phase of the second rifting, Figure 7c), all four segments were in the isolated growth stage, and the synthetic-approaching transfer zones were developed among these fault segments. During the Amal period (the post-rift of the second rifting, Figure 7b), fault $F_{3-3}$ and fault $F_{3-4}$ entered the soft-linkage stage, and the synthetic-overlapping transfer zone was developed between them; fault $F_{3-1}$ and fault $F_{3-2}$, and fault $F_{3-2}$ and fault $F_{3-3}$ began to connect initially at the top of the Bentiu Formation, indicating that the above two pairs of faults will soon enter the soft-linkage stage. During the Senna-Zeraf period (the third rifting, Figure 7a), four fault segments grew rapidly, with all entering the hard-linkage stage and eventually developing a fully grown, large fault.

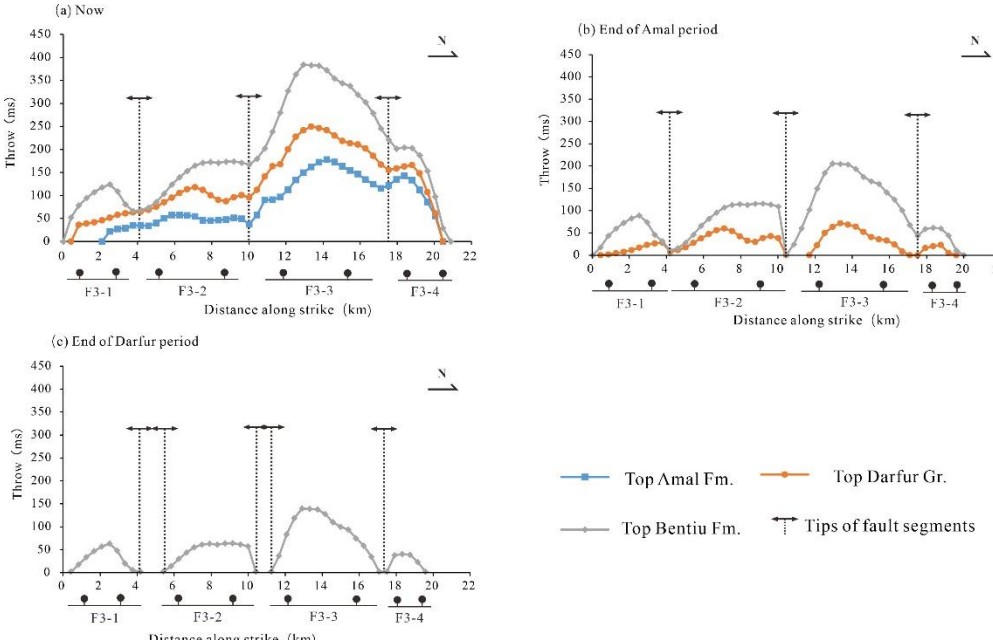

**Figure 7.** Relation between fault throw and extension distance of fault $F_3$. The presented plots depict the growth process of Fault $F_3$, demonstrating that the four individual fault segments comprising Fault $F_3$ became active during the Darfur period. Subsequently, two of them entered the soft-linkage stage during the Amal period and they all further advanced to the hard-linkage stage during the third rifting, eventually coalescing to form a fully grown, large fault.

Fault $F_2$ is segmented into three fault segments, from $F_{2-1}$, $F_{2-2}$ to $F_{2-3}$. Fault $F_{2-2}$, fault $F_{2-1}$, and fault $F_{2-3}$ have been active since the Bentiu period, the Darfur period, and the Amal period, respectively. Fault $F_{2-2}$ grew active during the Bentiu period (the post-rift phase of the first rifting, Figure 8d), while its activity intensity appeared to be weak with a maximum throw value of merely 10 ms; fault $F_{2-1}$ became active during the Darfur period (the syn-rift phase of the second rifting, Figure 8c), and was in the middle of an isolated growth, developing a synthetic-approaching transfer zone with fault $F_{2-2}$. During the Amal period (the post-rift phase of the second rifting, Figure 8b), fault $F_{2-3}$ became active, and it was then in isolated growth phase with fault $F_{2-2}$, while at the top of the Bentiu Formation, fault $F_{2-1}$ and fault $F_{2-2}$ became connected, marking the beginning of their soft-linkage phase and the development of a synthetic-overlapping transfer zone. These three fault segments saw rapid growth during the Senna-Zeraf period (the third rifting, Figure 8a), with all of them entering the hard-linkage phase, eventually giving rise to a fully grown, large fault.

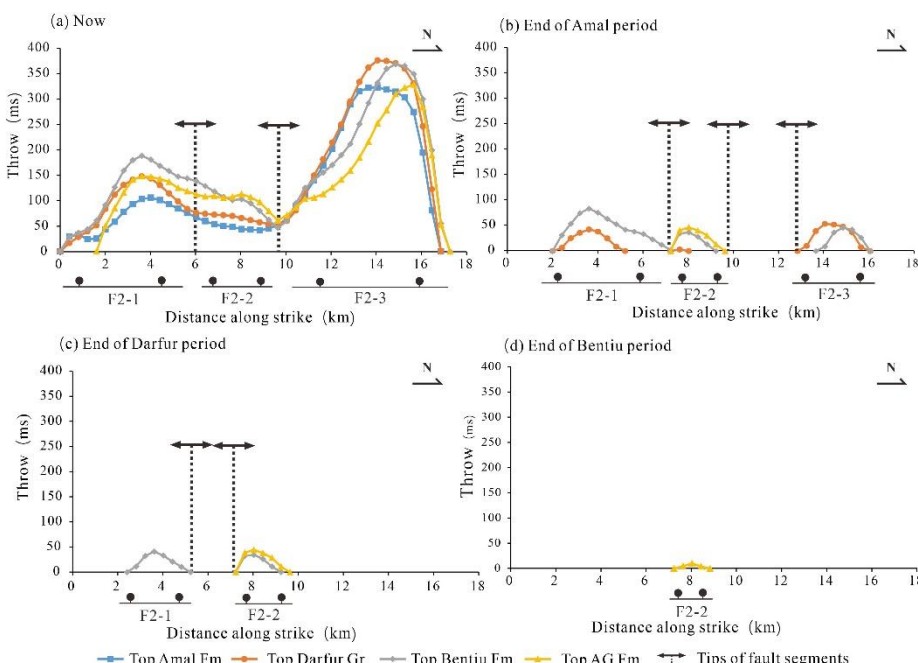

**Figure 8.** Relation between fault throw and extension distance of fault F$_2$. The provided diagrams illustrate the progressive growth of Fault F$_2$, showing that the three distinct fault segments forming Fault F$_2$ became active at different times, starting from the Bentiu period and continuing until the Amal period. Subsequently, two of these segments, F$_{2-1}$ and F$_{2-2}$, transitioned into the soft-linkage stage during the Amal period, and all segments further advanced to the hard-linkage stage during the third rifting, ultimately merging to form a fully developed, large fault.

Based on the typical profile data and the deposition time of each geological period from the previous research [23], we obtain the statistical map of the throw accumulation rate of the fault segments in each geological period, which allows us to quantify the activity intensity of these fault segments.

Comparing longitudinally the throw accumulation rates for the same profile at different times (Figure 9), we find that the weakest fault activity occurs during the first rifting. Only fault F$_{2-2}$ (Profile j in Figure 9) was then active, and the throw accumulation rate is just 0.2 ms/Myr The fault activity grew stronger in the second rifting, and the activity of all the fault segments in the three main faults grew very intense, with the average throw accumulation rate reaching 4.2 ms/Myr. Of particular note is fault F$_{1-2}$ (Profile b in Figure 9) in the south of fault F$_1$, which reached a rate of 8.4 ms/Myr. The third rifting saw a rapid drop in fault activity, with the average throw accumulation rate going down to merely 1.9 ms/Myr. Even for fault F$_{2-2}$ (Profile j in Figure 9), its throw accumulation rate is only 0.8 ms/Myr during this period. In general, the activity intensity of fault segments is featured by "extremely weak-strongest-weak" in these three rifting cycles.

Located in the south of the CFZ, the fault F$_{1-2}$ (Profile b in Figure 9) of fault F$_1$ was the most active during the second rifting, with its throw accumulation rate reaching 8.4 ms/Myr. The fault F$_{2-3}$ of fault F$_3$ (Profile k in Figure 9) was located in the north of the CFZ and emerged as the most active during the third rifting, with its throw accumulation rate reaching 5.7 ms/Myr. It can be seen that the extensional action within the CFZ was uneven on the plan, and rapid northward migration of the extensional center occurred from the second rifting to the third rifting.

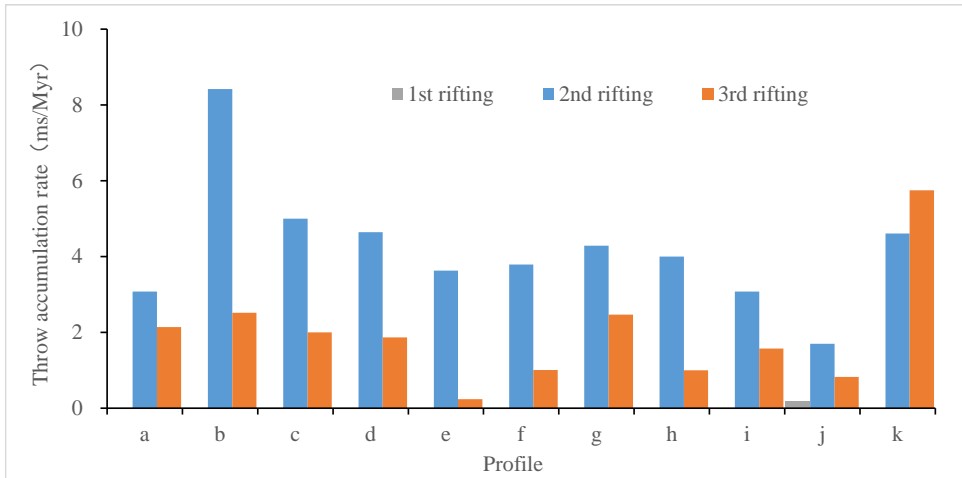

**Figure 9.** Statistical chart of throw accumulation rate of the CFZ. Locations of these profiles a–k are shown in Figure 4. The three rifting phases are represented in gray, blue, and orange. It is noteworthy that the main faults of the CFZ mainly became active during the second rifting. During the second rifting, the strongest fault activity was observed near the profile c of Fault $F_1$, which is the southern CFZ. In contrast, during the third rifting, the strongest fault activity was observed near the profile k of Fault $F_2$, which is the northern section of the CFZ.

Thus, the main faults comprising the CFZ have been active mainly since the second rifting. The fault segments that make up the main faults of the CFZ were in the isolated growth stage during the second syn-rift, partially entered the soft-linkage stage during the post-rift of the second rifting, and went through the hard-linkage stage and then eventually formed full-grown faults from the third rifting until now (Figure 10). The above faults weakened abruptly in activity intensity during the transition period of these two rifting cycles. The extensional center in the study area underwent a rapid northward migration during the last two rifting cycles.

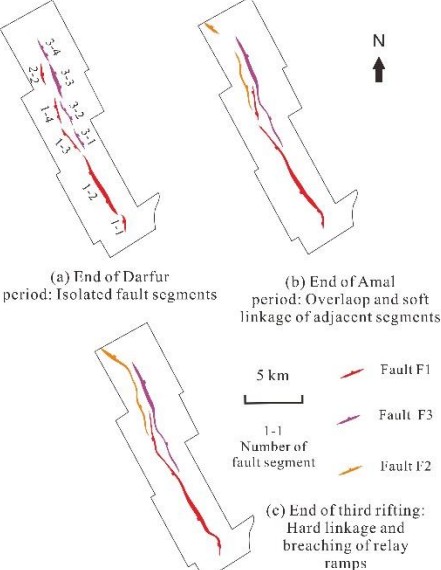

**Figure 10.** Evolution of the main faults by segment linkage on the top Bentiu Fm. in the CFZ. (**a**) Isolated fault segments grow from their tips during the second syn-rift. (**b**) Overlap of fault segments leads to soft linkage and the development of relay ramp structures during the post-rift of the second rifting. (**c**) Continued fault growth eventually leads to breaching and concealing of the relay ramps as the fault segments become hard-linked in the third rifting.

## 7. Discussion

### 7.1. Kinematics of the Study Area

The evolution of the Fula sag is subject to the break-up of the Atlantic Ocean, the northward movement of the Indian Plate, and the openness of the Red Sea.

The first rifting of the Fula sag was mainly influenced by Atlantic opening [45–48]. When the shear movement extended north-eastward out of the Congo Craton in the right-lateral strike-slip CASZ, the tips of the CASZ were changed into an extensional stress field, inducing the development of two NS-striking boundary faults in the Fula sag (Figure 1a faults ① and ②), and a near NS-tectonic striking (Figures 11a and 12a). During this period, the CFZ was not developed (Figure 9).

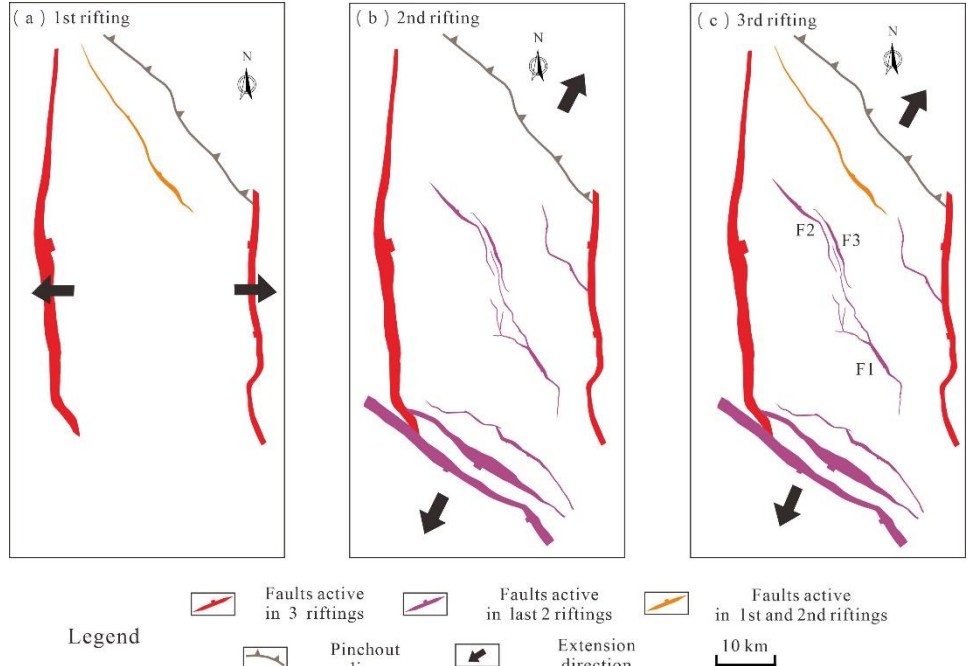

**Figure 11.** Evolution of the main faults in the Fula sag. The CFZ faults depicted in Figure 1b represent F1, F3, and F2 from south to north. The black arrows represent the extension direction during the corresponding periods. Noted that during the first rifting, the Fula sag experienced extension in the east-west direction, whereas in the subsequent two rifting phases, the extension direction shifted to the northeast direction.

The second rifting of the Fula sag is mainly influenced by the rapid opening of the Indian Ocean, which then created a near-NE-striking stress field in central Africa and caused the Fula sag to enter the second rifting [50,61,62]. Combined with the authors' previous study [20], it can be found that the second rifting not only caused the development of the southwestern fault-array-and-ramp zone and the CFZ in a near-perpendicular stress field (Figure 11b), but also sustained the activity of the east-west boundary faults (Figure 1a faults ① and ②). The tectonic pattern of the Fula sag's "overall NS-trending, internal tectonic NW-trending" was also finalized during this period (Figure 12b). The activity intensity of the CFZ was the strongest in the second rifting (Figure 9). The fault segments were in the isolated growth stage during the second syn-rift and partially entered the soft-linkage stage during the second post-rift.

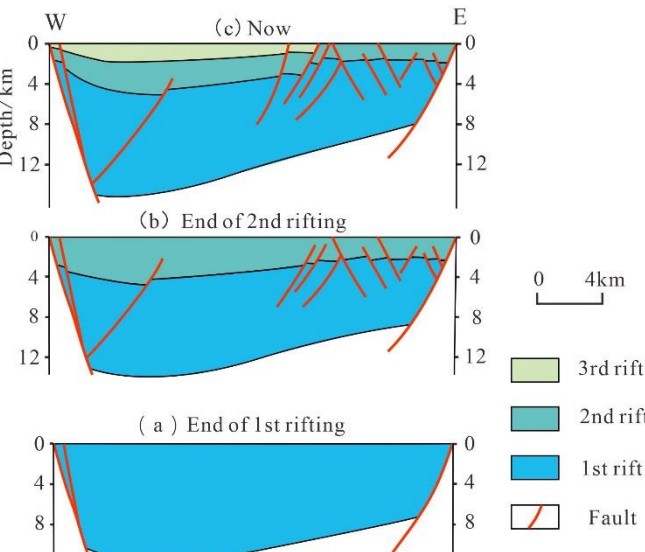

**Figure 12.** Development of muti-phase rifting in the Fula sag. It is noteworthy that the CFZ began to develop during the subsequent two rifting phases.

The third rifting of the Fula sag is mainly subject to the separation of the Red Sea. When the Red Sea Rift began to expand into the newborn ocean during the Pliocene [49], an east-to-west weakening, near-NE-striking extensional stress field was forming within the African plate [50,51]. Its main impact on the Fula sag is the formation of the main faults in the northeastern fault-array-and-ramp zone, while the rest of the main faults in the sag continued to be active (Figures 11c and 12c). The activity intensity of the CFZ was then the weakest (Figure 9). The fault segments here eventually formed full-grown faults during the third rifting.

In a nutshell, the first rifting was mainly influenced by the Atlantic Ocean opening, which led to the development of two boundary faults in the Fula sag and the Fula sag began to form a near north-south-trending tectonic pattern. The CFZ was not developed during this period. The second rifting is influenced by the opening of the Indian Ocean. The development of the southwestern fault-array-and-ramp zone and the CFZ began. The Fula sag, therefore, formed the tectonic pattern of "overall north-south-trending, internal tectonic north-west-trending". The activity intensity of the CFZ was then the strongest and the fault segments entered the soft-linkage stage. The third rifting is influenced by the separation of the Red Sea and the formation of the northeastern fault-array-and-ramp zone began. The activity intensity of the CFZ was then the weakest. The fault segments went through the hard-linkage stage and eventually formed full-grown faults. Finally, it should be noted that during the last two rifting cycles, the opening of the Fula sag under the NE direction introduces a left-lateral strike-slip kinematic regime to the NS-striking boundary faults.

### 7.2. Insights for Hydrocarbon Exploration

The lacustrine dark shale in AG Group the is the only mature source rock within the Fula sag, and all discovered oil and gas in the sag originate from this source rock in the southern sub-sag [63]. The AG group source rock in the southern sub-sag started expelling hydrocarbons during the second rifting period and reached its peak generation during the Paleogene (Li, W. et al., 2018) [64]. Oil and gas migrated and accumulated within traps along both sides of the main faults. Our study reveals that the central fault zone (CFZ) was more active during the second rifting, and oil and gas were trapped and accumulated in this area. During the third rifting, activity decreased, and most oil and gas remained below the Ghazal and Baraka group shales. Through the investigation of the growth history of the CFZ's main faults, 3, 3, 2 relay ramps have been identified on the fault planes of $F_1$, $F_3$, and

$F_2$, respectively. These relay ramps are potential locations for favorable sand bodies and lithological oil and gas reservoirs, and they represent promising areas for future lithological oil and gas exploration.

### 7.3. Study Limitations

One limitation of our analysis is the lack of an accurate regional velocity model, which restricts the conversion of the seismic data from time domain to depth domain and consequently affects the precision of the interpretation. Second, some of the fault segments exhibit asymmetric throw profiles, deviating from symmetric bell-shaped curves, towards the relay zones (e.g., $F_{1-2}$ and $F_{1-3}$, also in $F_3$ central segments; Figures 6 and 7), indicating that these fault segments are composed of smaller-scale segments, which appeared, grew, and interconnected during the Darfur period. However, this process could not be captured by the throw-distance plots at the end of the Darfur period.

Finally, certain issues in this study remain unresolved and require further investigation, such as understanding how the rapid opening during the second rifting led to greater rejections. Additionally, the substantial sediment thickness of up to 6 km during the first rifting suggests significant tectonic activity, yet the boundary faults of the Fula sag were the only ones displaying activity, while internal faults showed no activity. The underlying mechanism responsible for this phenomenon remains uncertain.

### 8. Conclusions

(1) The three rifting cycles of the Fula sag were subject to the segmental expansion of the Atlantic Ocean, the rapid opening of the Indian Ocean, and the separation of the Red Sea, respectively. During the first rifting, the boundary faults of the Fula sag began to develop, resulting in a nearly NS-trending tectonic pattern in the Fula sag; during the second rifting, the southwestern fault-array-and-ramp zone and the CFZ began to develop, forming the tectonic pattern of "overall NS-trending, internal tectonic NW-trending" in the Fula sag; during the third rifting, the northeastern fault-array-and-ramp zone began to form.

(2) The relative dip direction of the two faults involving $F_1$ and $F_3$ is divergent. The map position of the fault relative to the other is overlapping. The tectonic style in the middle section is a horst. Hence, we interpret the Fula-Moga transfer zone as the divergent overlapping type. The main faults $F_1$, $F_3$, and $F_2$ of the CFZ are laterally segmented into 4, 4, and 3 fault segments, respectively.

(3) The main faults comprising the CFZ have been active mainly since the second rifting. The fault segments that make up the main faults were in the isolated growth stage during the syn-rift period of the second rifting, entered the soft-linkage stage in the post-rift period of the second rifting, and went through the hard-linkage stage before finally forming a fully grown, large fault in the third rifting. The above faults weakened abruptly in activity intensity during the transition period between the last two rifting cycles.

**Author Contributions:** Conceptualization, Y.W. and G.Z.; methodology, Y.W.; validation, G.Z., G.W. and Z.C.; investigation, Y.W.; resources, R.R.; data curation, Y.Z. and K.G.; writing—original draft preparation, Y.W.; writing—review and editing, Y.W. and R.R.; supervision, Z.C. and Y.Z.; project administration, G.W.; funding acquisition, K.G. All authors have read and agreed to the published version of the manuscript.

**Funding:** The publication of the paper was funded by Science and technology project of China National Petroleum Corporation, grant number 2021DJ0301.

**Institutional Review Board Statement:** Not applicable.

**Informed Consent Statement:** Not applicable.

**Data Availability Statement:** Not applicable.

**Acknowledgments:** The research results of "Fine Exploration Field Evaluation and Target Selection in Muglad Basin, Sudan", a major scientific and technological project of China National Petroleum Corporation, were cited in the writing of this paper, and Jiguo Liu, Weili Ke, Quan Zou and other experts are gratefully acknowledged for assisting in data collection.

**Conflicts of Interest:** The funders had no role in the design of the study; in the collection, analyses, or interpretation of data; in the writing of the manuscript; or in the decision to publish the results.

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
