# Peer review of "Geometry and Kinematics of the Central Fault Zone, Fula Sag, Central Africa Shear Zone"

_applsci, doi:10.3390/app13169117_

Round 1
Reviewer 1 Report
See attached review report.

Moderate editing of English language and structure is needed. Details are listed on the review report.
Reviewer 2 Report
Dear Editor and Authors,
I appreciate the opportunity to revise the manuscript (applsci-2466856) titled “Geometry and kinematics of the central fault zone, Fula sag, Central Africa Shear Zone” by Wang et al. This manuscript uses 3D seismic data in order to provide a fault throw analysis and constrain an evolutionary model of fault growth in the Fula sag, Muglad Basin, a basin of the Central Africa Rift System. In general, the manuscript is well-written and well-illustrated, as well as deserves the interest of the reader. Nevertheless, the manuscript needs some improvements in its present form before its publication in Applied Sciences. Please find my review below, where I report some suggestions to improve this excellent manuscript. Furthermore, note that I have attached a PDF file with detailed revisions.
The first point is related to the abstract. The Abstract is not complete, very subjective, and uninformative. Important information is missing (not just in the abstract). For instance, at any moment, the abstract explains the methodology used in this study. In addition, several results described in the manuscript were not addressed in the abstract. For example, nothing about fault throw analysis, which is crucial to this study, is mentioned. Therefore, the abstract does not match what the manuscript shows.
The second point, and one of the main problems of this manuscript, is the absence of a methodology section explaining in detail the methods applied. For example, what is the coverage area of the seismic data? What is the spacing of the inlines and crosslines? What is the polarity of the data? What is the imaging depth? Are there wells in the study area? If yes, where are they? If so, were they used? How was the seismic interpretation performed? Which type of analysis was applied: fault throw or fault displacement? How was the modeling performed to analyze the fault throw/displacement? Why is the fault throw/displacement displayed in the time domain? What are the limitations of the applied methods? In addition, at no time is it explained which surface was considered to estimate the fault throw/displacement or if all surfaces were considered (as shown in Figure 4). Please, write a methodology section and be extremely detailed.
The third point comprises some sections poorly explored. For instance, the Geological Setting should provide complete support for the understanding of the basin rifting stages. For example, the paleostress field orientation of each rifting phase is shown only in Item 6 (Figure 11). The ideal would be to present a general contextualization of the Central Africa Rift System from a point of view of its tectonic origin. Also, the discussion should address the limitations of this study. For example, is there no velocity model, and so you chose to use time domain data? Have other studies done something similar? How might this impact your analysis?
However, my main question is the lack of a well-exposed scientific gap. It is not clear what is the gap addressed in this study. There is good tailings analysis that provides support for understanding fault growth in a portion of the basin analyzed. However, why is this relevant? Why will other readers read this work? This has a direct impact on the discussion of the work, which is little explored. Classic works on fault growth (e.g., Gawthorpe and Leeder 2000; and several others) and others that analyze fault tailings were not mentioned. Furthermore, the title of the manuscript suggests a kinematics analysis. However, this term is not mentioned in the discussion section.
Finally, because of all comments and other minor revisions in the attached PDF file, I recommend several modifications, which can be described as MAJOR REVISIONS.

Round 2
Reviewer 1 Report
Based on the authors' response letter and reading the manuscript again I believe that the manuscript has been sufficiently improved to warrant publication in Applied Sciences.
Minor editing in English is recommended for the final version of the manuscript. (For example, when referring to bookshelf faults depending on how you use it in the text it can be as ''bookshelf faults'' or ''forming a bookshelf-style of faulting''. Don't use it as ''the bookshelf faults''.
Minor editing in English is recommended for the final version of the manuscript.